# IGeoCAD: Unlocking Intrinsic Geometric Awareness for CAD Command Sequence Generation

## Abstract

The CAD command sequence generation task aims to translate design requirements into an executable sequence of precisely parametric commands for 3D asset construction, facing the critical challenge of capturing complex geometric relationships among primitives, especially points and curves. Previous methods attempt to enhance geometric awareness by differentiating commands from their parameters through parametric guidance. However, this simple distinction only implicitly models geometric awareness, failing to adequately represent and enforce crucial relationships at both the point and curve levels. In this paper, we contend that robust geometric reasoning requires a model to possess a dual awareness: point awareness to align with high-level design requirements, and curve awareness to maintain procedural consistency throughout the generation process. To this end, we introduce IGeoCAD, a framework that implements this dual awareness by explicitly modeling constraints through two simple yet effective modules: the Point Constraint-Aware Module (PCM) and the Curve Constraint-Aware Head (CCH). Specifically, at the point level, PCM interprets the geometry of each command, aligning high-level requirements with keypoints and modeling spatial dependencies through coupled attention, thereby ensuring precise localization and consistent point constraints. Concurrently, at the curve level, CCH fosters sequence geometric consistency by tracking dependencies across commands through a multi-task objective where an auxiliary head infers constraints to condition parameter prediction. Collectively, these modules significantly enhance IGeoCAD's geometric awareness, enabling the generation of complex and precise CAD programs. Extensive experiments demonstrate the effectiveness of the proposed approach.

## 1 Introduction

Computer-Aided Design (CAD) serves as the foundational technology for high-precision 3D modeling across various industries (Junk & Kuen, 2016; Cherng et al., 1998; Vido et al., 2024). With the accelerating paradigm shift toward intelligent manufacturing in recent years, the CAD command generation task has attracted increasing attention from both research and industry communities (Li et al., 2025; You et al., 2024). This task aims to automatically generate a software-executable sequence of CAD command types (non-parameters, e.g., lines, circles, extrusions) with corresponding parameters (e.g., start points, radii, distances) from design requirements (Wu et al., 2021).

A central challenge for CAD command generation is to accurately capture the complex geometric relations that define a 3D model. These relations are directly defined by constraints among basic primitives, especially points and curves. For instance, points are governed by explicit constraints like collinearity and coplanarity, and their overall arrangement reveals the design requirement. Similarly, curves exhibit critical geometric relationships, including parallelism, perpendicularity, and intersection, which precisely determine the final shape of the 3D asset. Although these geometric relationships are fundamental, most existing methods (Wu et al., 2021; Khan et al., 2024; Wang et al., 2025) primarily treat this task as a sequence generation problem, focusing on distinguishing parameters from their corresponding commands (non-parameter elements) by parametric guidance like loss, tokenization, or position encoding. As a result, they only implicitly model geometric awareness

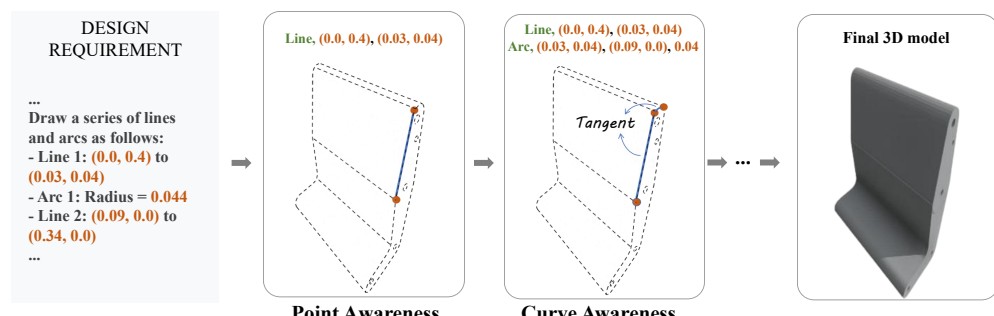

Figure 1: An illustration of the CAD modeling process and dual geometric awareness for CAD command generation. Point awareness maps high-level design requirements to precise point parameters for accurate point localization. In parallel, curve awareness models the dependencies between sequential commands (e.g., a tangent constraint between a line and an arc) to maintain global geometric integrity. This dual awareness enables the generation of geometrically coherent 3D models.

through token-level optimization, failing to adequately represent and enforce crucial relationships at both the point and curve levels.

To address these limitations, we argue that a model must explicitly learn geometric constraints at the point level (Zhou et al., 2021) and the curve level (Nash et al., 2020). We observe that point-level constraints are primarily driven by high-level design requirements, whereas curve-level constraints stem from the procedural logic of the command sequence, as shown in Figure 1. We refer to these two abilities as point awareness and curve awareness. Achieving point awareness requires the model to map high-level design intent onto precise point parameters and reason about their spatial interdependencies. This is crucial for accurate entity localization while maintaining foundational geometric consistency. In contrast, curve awareness entails explicitly tracking the geometric dependencies induced by successive commands. This is essential for preventing command-level conflicts and preserving global geometric integrity.

To this end, we propose IGeoCAD, an innovative framework designed to unlock the model's intrinsic geometric awareness by leveraging point constraints and curve constraints. IGeoCAD encodes requirement-driven constraints on points and dependencies along command sequences, guiding the model to produce consistent CAD assets. To achieve this, we devise two simple yet effective plug-and-play modules: the Point Constraint-Aware Module (PCM) and the Curve Constraint-Aware Head (CCH). At the point level, the PCM aligns high-level requirements with the relevant geometric keypoints, and models spatial dependencies among points through a coupled attention mechanism (Vaswani et al., 2017). This enables precise localization and consistent point constraints. In parallel, at the curve level, the CCH maintains consistency across commands by a multi-task learning (Zhang & Yang, 2021). This guidance helps the 3D asset avoid conflicts and preserve global geometric integrity. Collectively, PCM and CCH enhance point awareness and curve awareness within IGeoCAD, resulting in more complex and precise CAD assets. Our main contributions are summarized as follows:

- We systematically dissect geometric awareness for CAD command generation into two distinct levels: point awareness and curve awareness. We identify that point constraints are primarily dictated by high-level design requirements, while curve constraints emerge from the procedural logic of the command sequence.

- We propose **IGeoCAD**, a novel framework designed to explicitly enhance geometric awareness. IGeoCAD features two simple yet effective modules: the Point Constraint-Aware Module (PCM) to capture requirement-driven point constraints through a coupled attention mechanism, and the Curve Constraint-Aware Head (CCH) to enforce procedural curve constraints via multi-task learning.

- We introduce two novel metrics designed to evaluate geometric constraint satisfaction at both the point and curve levels. We conduct extensive experiments to demonstrate the effectiveness of IGeoCAD.

## 2 RELATED WORK

### 2.1 END-TO-END 3D GENERATION

Recent advancements in generative models have enabled the direct synthesis of 3D assets from multimodal user prompts. These state-of-the-art methods can produce various representations, including point clouds (Zhou et al., 2021; Nichol et al., 2022), polygonal meshes (Wang et al., 2024; Groueix et al., 2018), and Boundary Representations (B-Reps) (Sharma et al., 2020; Wang et al., 2020; Xu et al., 2024b). However, their end-to-end nature often results in a "black-box" process, which significantly limits both interpretability and editability.

### 2.2 CAD COMMAND SEQUENCE GENERATION

Distinct from end-to-end 3D generation, CAD command sequence generation is a procedural paradigm that offers superior interpretability and editability (Wu et al., 2024; Zhou et al., 2023; Xie & Ju, 2025; McCarthy et al., 2025), and is increasingly becoming a mainstream approach.

While DeepCAD (Wu et al., 2021) introduced CAD command sequence generation for 3D reconstruction, most follow-up research has concentrated on conditional generation from various inputs, including geometric data like point clouds or B-Reps (Zhou et al., 2023; Li et al., 2023; Uy et al., 2022), visual cues like images (Hong et al., 2024), and, most notably, semantic information from text descriptions (Khan et al., 2024). More recently, research has advanced towards multimodal systems capable of processing a combination of these inputs (Wang et al., 2025; Xu et al., 2024a; Kolodiazhnyi et al., 2025; Wu et al., 2025).

### 2.3 GEOMETRIC AWARENESS

Efforts to enhance geometric awareness in CAD sequence generation generally follow two directions. The first relies on extrinsic guidance that separates command types from parameters, using distinct losses or rewards (Guan et al., 2025; Wu et al., 2021) or separate token domains and specialized encodings (Khan et al., 2024; Wang et al., 2025). This separation often reduces parameters to abstract numeric tokens and overlooks their intrinsic geometric semantics, limiting geometric reasoning. The second augments architectures to inject geometric priors more directly, including dual-branch designs for commands and parameters (Seff et al., 2021), decoupled generation of entities and constraints for local consistency (Seff et al., 2020; Para et al., 2021; Willis et al., 2021), and hierarchy-aware or grammar-based formulations for global topology (Xu et al., 2022; Para et al., 2021; Zhang et al., 2024; Li et al., 2025). While these strategies improve focus on geometric relations, they introduce redundancy and complicate training. In contrast, our unified framework adopts two plug-and-play modules that explicitly model point-level and curve-level constraints, strengthening geometric coherence.

## 3 METHODS

### 3.1 TASK DEFINITION

The primary objective of the CAD command sequence generation task is to generate a software-executable, parametric CAD command sequence $\mathcal{S}$ that constructs a 3D model based on given design requirements $\mathcal{R}$. The sequence $\mathcal{S} = [S_1, S_2, \ldots, S_N]$ is an ordered list of $N$ commands. Following the standard autoregressive formulation, this task models the conditional distribution over sequences as:

$$P(S|R) = \prod_{i=1}^{N}(S_i|S_{1:i-1}, R). \tag{1}$$

Each command $S_i = (t_i, p_i)$ comprises:

- A command type $t_i$ selected from a predefined vocabulary $\mathcal{T}$ (e.g., Circle, Extrusion).
- A corresponding parameter set $\mathbf{p}_i$ that typically includes point parameters, and operational arguments (e.g., extrusion distance and direction).

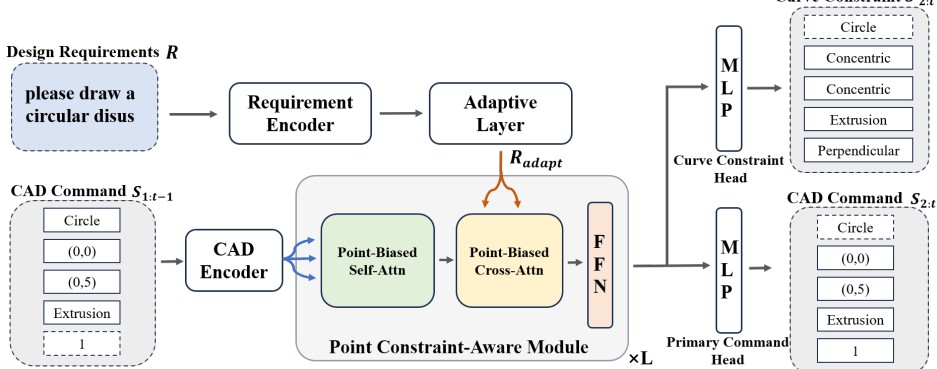

Figure 2: IGeoCAD Architecture. IGeoCAD takes design requirements $R$ as input, which are processed by an encoder to produce an adaptive embedding $\mathbf{R}_{\text{adapt}}$. This embedding conditions the $L$ layer autoregressive decoder. At each decoding step, the Point Constraint Aware Module enhances the representation of geometric entities through point-biased self-attention and cross-attention. Finally, two parallel heads, the Command Prediction Head and the Curve Constraint-Aware Head, jointly predict the next CAD command and its associated geometric constraints, respectively, to ensure the coherence and validity of the generated design.

## 3.2 FRAMEWORK OVERVIEW

As shown in Figure 2, IGeoCAD uses a standard encoder-decoder Transformer architecture. The encoder maps design requirements $R$ to an alignment embedding $\mathbf{R}_{\text{adapt}}$ that conditions the decoding process. The $L$ layer decoder autoregressively predicts tokens, and its final hidden representations are then passed to the Command Prediction Head, a Multi Layer Perceptron (MLP), to produce the CAD commands sequence $\mathcal{S}$.

Beyond this backbone, we introduce two geometry-aware modules. First, the Point Constraint-Aware Module enhances attention to point tokens within the decoder layers, guiding the model to focus on precise point locations and thereby enhancing its understanding of geometric boundaries and the global structure of 3D assets. Second, the Curve Constraint-Aware head operates in parallel with the final MLP, explicitly reasoning over command-level curve constraints during generation to maintain contextual coherence and geometric validity.

## 3.3 POINT CONSTRAINT-AWARE MODULE

CAD command boundaries and trajectories are defined by key points (e.g., Line: start point/end point; Circle: center point/top point). Accurate localization of these points is essential for capturing geometric boundaries and object structure. Conventional models often treat point parameters as ordinary tokens, overlooking their geometric interdependencies.

We introduce the Point Constraint-Aware Module (PCM) to explicitly encode point-level relationships. PCM employs a Point-Bias Attention Mask in both self-attention and cross-attention in the decoder layers to enhance point awareness, which strengthens point-to-point correlations and point-to-requirement constraints and thereby improves geometric awareness.

### 3.3.1 POINT-BIAS ATTENTION MASK

The Point-Bias Attention Mask is the core mechanism of our Point Constraint-Aware Module (PCM). It embeds geometric priors into the attention computation by selectively amplifying attention weights for point tokens within a CAD command sequence. Concretely, attention is computed as:

$$A = \text{Softmax}(M_{pb}(\frac{QK^T}{\sqrt{d_k}}))V, \tag{2}$$

where $Q$, $K$, and $V$ are the query, key, and value matrices, $d_k$ is the key dimensionality, and $A$ is the resulting attention output. The mask $M_{pb}$ adds a bias mask to point tokens, thereby enhancing geometric relationships among points.

We integrate this Point-Bias Attention Mask into the Transformer decoder's self-attention and cross-attention layers, forming two specialized components: the Point-Biased Self-Attention and the Point-Biased Cross-Attention. This combined approach significantly improves the model's point-level geometric awareness.

### 3.4 ARCHITECTURAL INTEGRATION

Figure 3: Toy example of the Point-Biased Attention Mask. Left: Point-Biased Self-Attention. Columns 2 and 3 are point tokens; the mask boosts attention from all query tokens to these columns, enforcing point-to-point consistency. Right: Point-Biased Cross-Attention. Rows 2 and 3 are point tokens; the mask boosts attention from these rows to all requirement tokens, ensuring point-to-requirement alignment. Darker cells indicate stronger bias.

**Point-Biased Self-Attention.** We implement Point-Biased Self-Attention by inserting a Point-Bias Mask into the decoder self-attention layers. As shown in Figure 3, the Point-Biased Self-Attention Mask strengthens attention from each query token to point tokens in the key. In this way, each new parameter is conditioned on the geometric structure formed by earlier points. This design preserves point-point geometric consistency during autoregressive decoding.

**Point-Biased Cross-Attention.** We implement Point-Biased Cross-Attention by inserting a Point-Bias Mask into the cross-attention layers. As shown in Figure 3, the Point-Biased Cross-Attention Mask strengthens attention from point tokens in the query to the relevant design requirement tokens in the key. In this way, each new point is conditioned on the global layout specified by the requirements. This design promotes point-requirement coherence during autoregressive decoding and produces assemblies that follow high-level specifications while remaining geometric coherence at the point level.

### 3.5 CURVE CONSTRAINT-AWARE HEAD

Ensuring geometric and topological consistency across command sequences is central to CAD command generation. Each new geometric command must satisfy relational constraints induced by prior commands, such as tangency, perpendicularity, and parallelism.

We propose a Curve Constraint-Aware Head (CCH) that explicitly models dependencies among sequential curves. To train CCH, we additionally construct a Curve Constraint Sequence to provide effective supervision over curve-level constraints, and adopt a dual-head training scheme to jointly optimize command generation and constraint satisfaction.

#### 3.5.1 CURVE CONSTRAINTS SEQUENCE

Standard CAD command sequence datasets (Wu et al., 2021) only provide raw commands but lack explicit geometric constraint information between these commands. To address this deficiency, we leverage the advanced reasoning and geometric inference capabilities of Large Language Models (LLMs) to generate Curve Constraints Sequence. Specifically, we employ the Gemini 2.5 Flash (Comanici et al., 2025) model to parse long command sequences to infer the associated geometric constraints between adjacent commands.

Although curve constraints can be modeled as a fully connected graph (Seff et al., 2020), this misaligns with CAD command generation by imposing too many command-to-command constraints, which raises complexity. Therefore, we simplify this graph structure, retaining only the constraint relationships between adjacent geometric constraints in the sequence. This simplification effectively linearizes the constraint information, allowing it to be represented in a sequential format.

Given a design requirement $R$ and an original command sequence $\mathcal{S}$, we use a large language model to infer a Curve Constraint Sequence $\mathcal{S}'$. Under a system prompt, the LLM outputs $\mathcal{S}' = [S'_1, S'_2, \ldots, S'_N]$, where each $S'_i = (t_i, c_i)$ encodes a discrete constraint step in the command generation process. The $S'_i$ comprises a command type $t_i$ and its associated geometric constraint $c_i$, where $t_i$ preserves the $i$-th command type present in $\mathcal{S}$, and $c_i$ specifies the geometric relationship between the $t_i$ and $t_{i-1}$. For $i = 1$, $c_1$ defines the relationship between the first command $t_1$ and the origin of the design space. The constraint type for $c_i$ is selected from a fixed vocabulary: Coincident, Perpendicular, Parallel, Tangent, Concentric, Intersectant, and Disjoint.

### 3.6 DUAL-HEAD TRAINING SCHEME

Building upon the Curve Constraints Sequence, we employ a Dual-Head Training Scheme that explicitly injects curve awareness into the generative pipeline, as shown in Figure 2. The Command Prediction Head handles the primary autoregressive task of generating the CAD command sequence $\mathcal{S}$. In parallel, the Curve-Constraint-Aware Head is trained with the Curve Constraint Sequence $\mathcal{S}'$ as supervision and is formulated as a classification objective. The overall training objective for IGeoCAD sums the losses from both heads:

$$\mathcal{L}_{\text{overall}} = \mathcal{L}_{\text{cad}} + \mathcal{L}_{\text{constraint}}, \tag{3}$$

where $\mathcal{L}_{\text{cad}}$ denotes the cross-entropy loss for CAD command generation, and $\mathcal{L}_{\text{constraint}}$ is the cross-entropy loss for classifying curve-constraint labels derived from $\mathbf{S}'$.

This dual-head design enables the model to explicitly understand the curve-level geometric meaning embedded in the CAD command. This constraint prediction serves as a powerful geometric regularization, leading to a coherent and geometrically accurate command sequence.

## 4 EXPERIMENT

### 4.1 EXPERIMENTAL SETUP

**Dataset** All experiments use the Text2CAD dataset (Khan et al., 2024), which pairs CAD command sequences with textual descriptions at four levels of granularity from L0 to L3. The L0 and L1 descriptions capture the global product shape and structural characteristics, whereas the L2 and L3 descriptions specify the stepwise modeling operations.

**Comparative Methods** Several recent research accept text to generate CAD command sequences (Wang et al., 2025; Xu et al., 2024a), but their models or data are not open source and thus unavailable for direct comparison. Consequently, our performance comparisons focus on Deep-CAD (Wu et al., 2021) and Text2CAD (Khan et al., 2024). We further benchmark mainstream large language models with strong text generation capabilities, including GPT-5 (2025-08-07) (OpenAI, 2024), Gemini 2.5 Flash (Comanici et al., 2025), and Qwen3-235B-A22B Instruct-2507 (Yang et al., 2025). We adopt a one-shot prompting setup: each model receives explicit CAD command rules, a single illustrative example, and precise design requirements, and is instructed to directly output the CAD command sequence, with more details are shown in the appendix.

**Metric** We evaluate the model along two axes: sequence completeness and geometric awareness. For sequence completeness, we use the F1 score (Khan et al., 2024). After aligning the predicted and reference command sequences using the Hungarian algorithm, we compute F1 scores for four primary command types: line, arc, circle, and extrusion. Higher F1 indicates stronger logical awareness in predicting command type. For geometric fidelity, we use two metrics. Chamfer Distance (CD) measures the distance between sampled points, and Jensen–Shannon Divergence (JSD) (Xu et al., 2023) quantifies the difference between their empirical distributions. Lower CD or JSD implies more accu-

rate command parameters, reflecting stronger geometric awareness. Together, these metrics provide a comprehensive evaluation of our method.

Moreover, we propose two new metrics to evaluate geometric awareness in CAD generation: Relative Position Consistency (RPC) and Geometric Constraint Consistency (GCC). RPC assesses point-level constraints by testing whether the model maintains original point-to-point relationships while remaining invariant to global translation; it computes the average discrepancy between corresponding relative vectors, and lower values indicate better preservation of internal shape structure. GCC evaluates curve-level constraints by using a large language model to extract constraint sequences from both the generated and the ground-truth CAD command sequence and then measuring their agreement; higher values indicate stronger adherence to geometric constraints at the curve level. More details are illustrated in the appendix.

## 4.2 COMPARISON WITH STATE-OF-THE-ART METHODS

Table 1: Quantitative evaluation of our method against state-of-the-art (SOTA) methods. The performance is assessed on four levels of text granularity (L0-L3) requirements ($\mathbf{R}$). Both the F1 score and CD are reported as average scores, with CD values multiplied by $10^3$. Bold indicates the best performance.

| Method | Text | Sequence Completeness | | | | Geometric Awareness | |
|---|---|---|---|---|---|---|---|
| | | F1↑ | | | | CD ($\times 10^3$) ↓ | JSD↓ |
| | | Line | Arc | Circle | Extrusion | | |
| *— LLM based Methods —* | | | | | | | |
| GPT-5 | L0 | 65.19 | 8.33 | 48.95 | 83.79 | 234.92 | 0.891 |
| | L1 | 74.10 | 0.00 | 61.40 | **94.91** | 245.41 | 0.869 |
| | L2 | 73.75 | 20.50 | 62.52 | 92.89 | 231.95 | 0.818 |
| | L3 | 82.54 | 26.80 | 78.63 | **95.36** | 65.69 | 0.300 |
| Qwen-3 | L0 | **72.34** | 0.00 | 53.39 | **88.75** | 220.58 | 0.867 |
| | L1 | **74.67** | 0.00 | 62.80 | 92.54 | 245.43 | 0.898 |
| | L2 | 74.91 | 7.78 | 71.14 | **94.17** | 160.70 | 0.759 |
| | L3 | 82.40 | **42.59** | **82.60** | 93.93 | 44.78 | 0.326 |
| Gemini-2.5 | L0 | 59.57 | 0.00 | **53.72** | 80.98 | 256.68 | 0.883 |
| | L1 | 73.01 | 0.00 | **65.14** | 92.10 | 282.66 | 0.903 |
| | L2 | 75.37 | 0.00 | 65.96 | 91.44 | 200.46 | 0.777 |
| | L3 | 62.48 | 2.08 | 53.45 | 81.16 | 43.59 | 0.329 |
| *— Non-LLM based Methods —* | | | | | | | |
| DeepCAD | L3 | 76.78 | 20.04 | 65.14 | 88.72 | 97.93 | – |
| Text2CAD | L0 | 65.38 | 3.94 | 50.10 | 85.43 | 233.87 | 0.824 |
| | L1 | 69.78 | 3.36 | 58.01 | 91.75 | 244.79 | 0.853 |
| | L2 | 72.71 | 6.69 | 66.06 | 93.39 | 150.11 | 0.708 |
| | L3 | 81.03 | 35.45 | 74.49 | 93.35 | 29.30 | 0.362 |
| **IGeoCAD (Ours)** | L0 | 67.86 | **5.65** | 52.34 | 86.77 | **180.24(-23%)** | **0.757(-8%)** |
| | L1 | 71.12 | **5.81** | 62.37 | **92.39** | **200.32(-18%)** | **0.732(-14%)** |
| | L2 | **75.67** | **11.75** | **71.63** | 93.42 | **132.36(-12%)** | **0.665(-6%)** |
| | L3 | **83.78** | 38.47 | 79.20 | 94.33 | **21.16(-28%)** | **0.243(-33%)** |

As summarized in Table 1, IGeoCAD consistently outperforms all competing methods across four levels of text granularity (L0 to L3) compared with non-LLM-based methods. It achieves an average 20% reduction in CD relative to the strongest baseline, Text2CAD, while simultaneously delivering higher F1 scores for all primitive categories across all granularity levels. The advantage is most evident under abstract L0 level requirements, where IGeoCAD reduces CD by 23% compared to Text2CAD (180.24 vs. 233.87), demonstrating robust inference of geometric intent from ambiguous instructions. Notably, even under expert-level L3 prompts with minimal ambiguity, IGeoCAD main-

tains a clear lead, attaining a 28% lower CD (21.16 vs. 29.30). Consistent gains are also observed at intermediate levels, with an 18% reduction on L1 and a 12% reduction on L2. Collectively, these results indicate that IGeoCAD reliably produces geometrically coherent CAD models across the full spectrum of prompt specificity.

We further compare IGeoCAD with mainstream LLMs. While LLMs demonstrate high command-level accuracy and competitive F1 scores, they exhibit significant deficiencies in geometric fidelity, evidenced by a higher CD. This observation highlights a key insight: proficiency in generating syntactically correct command sequences does not guarantee geometric precision. The gap comes from unreliable numeric grounding in parameter assignment. The LLMs often produce small errors in the numeric values that control dimensions and positions. These small errors accumulate and propagate through the command sequence, which raises the CD and harms the quality of the final 3D assets.

### 4.3 ABLATION STUDY

#### 4.3.1 QUALITATIVE ANALYSIS

Our qualitative ablation in Figure 8 shows that the baseline accumulates errors over the sequence, leading to missing parts and broken structures. The Point Constraint-Aware Module (PCM) explicitly models point-level coherence within each command, keeping geometric keypoints consistent and reducing local error. The Curve Constraint-Aware Head (CCH) enforces geometric consistency across commands, preserving relations between successive curve entities and guiding accurate parameter prediction for curve-level coherence. Together, PCM and CCH enable the IGeoCAD to produce shapes that are structurally complete and parametrically accurate, which improves fidelity and robustness. Additional visualizations and qualitative analyses are provided in the Appendix.

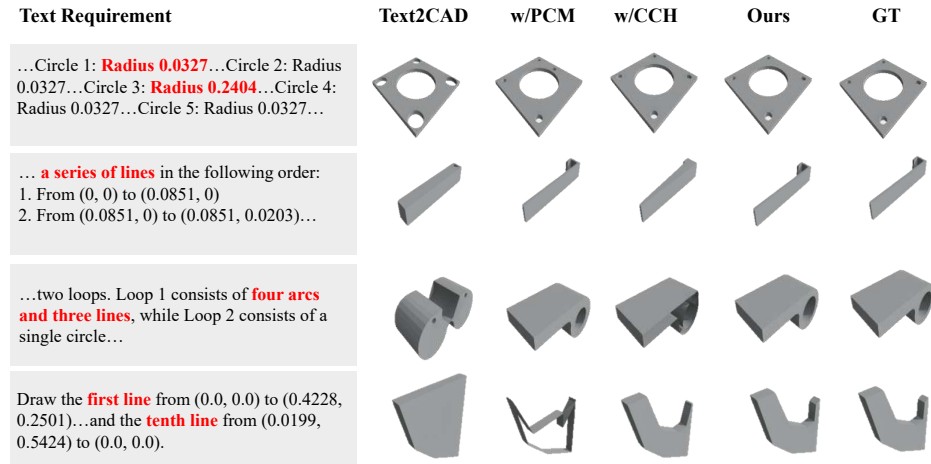

Figure 4: Qualitative ablation of IGeoCAD. PCM and CCH jointly reduce error.

#### 4.3.2 QUANTITATIVE ANALYSIS

To quantify the contribution of each module, we conduct a controlled ablation study. Results in Table 2 show that both the Point Constraint-Aware Module (PCM) and the Curve Constraint-Aware Head (CCH) individually outperform the baseline. When combined in the full model, IGeoCAD, they achieve the best results across all prompt granularity levels. These findings indicate a clear functional specialization between the two modules.

**PCM:** The coupled attention PCM is most effective at levels L0 to L2, which use short prompts with concise geometric instructions. This suggests that PCM excels at modeling point-level coherence among geometric keypoints and their dependencies, which is crucial when explicit textual guidance is limited.

**CCH:** The CCH is particularly effective at level L3, which contains long, descriptive prompts. It enforces geometric consistency across commands by predicting constraints between adjacent curves,

thereby guiding accurate parameter generation for curve-level coherence. This capability becomes more important as prompts grow in complexity and detail.

In summary, the two modules are complementary. PCM focuses on point-level keypoint coherence, while CCH enforces curve-level geometric consistency and guides parameter prediction. Together, they provide a robust and versatile understanding of geometric instructions, enabling consistent performance across the full range of prompt granularities.

Table 2: The quantitative ablation study of IGeoCAD. Performance is measured by F1 score and CD (multiplied by $10^3$) across four text granularity levels (L0-L3). Abbreviations are as follows: PCM (Point Constraint-Aware Module), and CCH (Curve Constraint-Aware Head).

| Method | Text | Sequence Completeness | | | | Geometric Awareness | |
|---|---|---|---|---|---|---|---|
| | | F1↑ | | | | CD $(\times 10^3)$ ↓ | JSD↓ |
| | | Line | Arc | Circle | Extrusion | | |
| w/o PCM w/o CCH | L0 | 65.38 | 3.94 | 50.10 | 85.43 | 233.87 | 0.824 |
| | L1 | 69.78 | 3.36 | 58.01 | 91.75 | 244.79 | 0.853 |
| | L2 | 72.71 | 6.69 | 66.06 | 93.39 | 150.11 | 0.708 |
| | L3 | 81.03 | 35.45 | 74.49 | 93.35 | 29.30 | 0.362 |
| w/o CCH | L0 | 65.97 | 4.55 | 50.43 | 85.60 | 193.42(-17%) | 0.775(-6%) |
| | L1 | 67.33 | 3.44 | 52.68 | 90.67 | 217.22(-11%) | 0.752(-12%) |
| | L2 | 73.42 | 9.87 | 69.10 | 93.11 | 148.76(-1%) | 0.677 (-4%) |
| | L3 | 82.98 | 27.67 | 78.01 | 94.02 | 25.71 (-12%) | 0.279 (-23%) |
| IGeoCAD (Ours) | L0 | **67.86** | **5.65** | **52.34** | **86.77** | **180.24(-23%)** | **0.757(-8%)** |
| | L1 | **71.12** | **5.81** | **62.37** | **92.39** | **200.32(-18%)** | **0.732(-14%)** |
| | L2 | **75.67** | **11.75** | **71.63** | **93.42** | **132.36(-12%)** | **0.665(-6%)** |
| | L3 | **83.78** | **38.47** | **79.20** | **94.33** | **21.16(-28%)** | **0.243(-33%)** |

As shown in Table 3 and Table 4, the Point Correlation Module improved the RPC metric by a large margin, while the Command Correlation Head brought only minor gains on RPC. This pattern indicates that the Point Correlation Module strengthens the model's ability to capture geometric relations at the level of individual points, which is consistent with its design goal. In contrast, the Command Correlation Head produced a substantial improvement in the GCC metric, whereas the Point Correlation Module had a negligible effect on GCC. This outcome confirms that the Command Correlation Head improves the model's understanding of geometric relations across commands, matching its intended design.

Table 3: Ablation Study of the PCM Module based on the RPC (Point-level) Metric. Lower values indicate better.

| | L0 | L1 | L2 | L3 |
|---|---|---|---|---|
| Baseline | 0.682 | 0.741 | 0.610 | 0.380 |
| w/ PCM | 0.632 | 0.604 | 0.479 | 0.349 |
| w/ PCM&CCH | **0.603** | **0.575** | **0.454** | **0.330** |

Table 4: Ablation Study of the CCH Module based on the GCC (Curve-level) Metric. Higher values indicate better.

| | L0 | L1 | L2 | L3 |
|---|---|---|---|---|
| Baseline | 0.642 | 0.687 | 0.723 | 0.796 |
| w/ CCH | 0.693 | 0.735 | 0.794 | 0.864 |
| w/ CCH&PCM | **0.697** | **0.744** | **0.802** | **0.873** |

## 5 CONCLUSION

We present IGeoCAD, a framework that enhances intrinsic geometric awareness by leveraging point and curve constraints. IGeoCAD explicitly models these constraints at the point and curve levels via two simple, plug-and-play modules: the Point Constraint-Aware Module (PCM) and the Curve Constraint-Aware Head (CCH). Together, PCM and CCH strengthen geometric reasoning, enabling the generation of more complex and precise CAD assets, as confirmed across prompts of varying complexity. The plug-and-play design of PCM and CCH facilitates transfer to diverse input modalities and model architectures.

## 6  ETHICS STATEMENT

This work adheres to the ICLR Code of Ethics. In this study, no human subjects or animal experimentation were involved. All datasets used, including Text2CAD datasets, were sourced in compliance with relevant usage guidelines, ensuring no violation of privacy. We have taken care to avoid any biases or discriminatory outcomes in our research process. No personally identifiable information was used, and no experiments were conducted that could raise privacy or security concerns. We are committed to maintaining transparency and integrity throughout the research process.

## 7  REPRODUCIBILITY STATEMENT

We have made every effort to ensure that the results presented in this paper are reproducible. All code and datasets have been made publicly available in an anonymous repository to facilitate replication and verification. The experimental setup, including training steps, model configurations, and hardware details, is described in detail in the paper, to assist others in reproducing our experiments.

Additionally, Text2CAD dataset is publicly available, ensuring consistent and reproducible evaluation results.

We believe these measures will enable other researchers to reproduce our work and further advance the field.

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
