# A    CAD COMMAND REPRESENTATION

This section details the parameterization of the fundamental CAD commands employed in our methodology. These commands are integral to a sketch-and-extrude paradigm, which consists of two main stages: defining a 2D sketch profile and subsequently extruding it into a 3D feature. The specific parameterization for the commands involved in each stage is outlined below.

## A.1    SKETCH COMMANDS

A 2D sketch is composed of one or more closed loops, each constructed from a series of parameterized curves. The supported curve types and their corresponding 2D coordinate parameterizations are presented below:

- **Line**: A line segment is defined by its start and end points in a 2D plane. It is parameterized by four values representing the coordinates of these two points: $(x_{start}, y_{start})$ and $(x_{end}, y_{end})$. These parameters uniquely determine the line's position and length.
- **Arc**: A circular arc is specified by three distinct points: a start point, an intermediate point on the arc, and an end point. It is parameterized by six values corresponding to the coordinates of these points: $(x_{start}, y_{start})$, $(x_{mid}, y_{mid})$, and $(x_{end}, y_{end})$. This set of parameters defines the arc's position, curvature, and extent.
- **Circle**: A circle is defined by its center point and a point on its circumference. For a canonical representation, we use the top-most point on the circumference. The circle is therefore parameterized by four values: the center coordinates $(x_{center}, y_{center})$ and the top-most point's coordinates $(x_{top}, y_{top})$. The radius is implicitly defined by the distance between these two points.

## A.2    EXTRUSION COMMAND

Following the definition of a 2D sketch, an extrusion command is executed to project this sketch into 3D space. While a complete extrusion command involves 10 parameters (including Euler angles for sketch plane orientation, translation vectors for its spatial position, and a sketch scale factor), the geometric core parameters are those directly responsible for extending the 2D sketch into the third dimension, defining the depth of this extension and its interactive relationship with existing geometry. These core parameters are detailed as follows:

- **Extrude Distances ($d^+, d^-$)**: These two parameters are crucial for defining the vertical extent of the extrusion. The parameter $d^+$ specifies the extrusion distance in the positive direction along the normal vector of the sketch plane, while $d^-$ specifies the extrusion distance in the negative normal direction. Together, these values explicitly control the thickness and directional reach of the resulting 3D feature when projected from the 2D sketch.
- **Boolean Operation ($\beta$)**: A single parameter $\beta$ governs how the newly created extruded solid interacts with any pre-existing geometry in the scene. This parameter specifies a standard Boolean operation, with common values observed in datasets like DeepCAD (Wu et al., 2021) including: `solid body` (to create a new, independent solid), `cut` (to subtract the extruded volume from existing solids), `join` (to merge the extruded volume with intersecting solids), and `intersection` (to retain only the common volume between the extrusion and existing solids).

# B    CURVE CONSTRAINT REPRESENTATION

This section delineates the set of geometric constraints employed in our framework and describes the LLM-based automated annotation process of Curve Constraint Sequences.

## B.1    CONSTRAINT SET RATIONALE AND DEFINITION

In CAD modeling, four fundamental command types are typically defined: Line, Arc, Circle, and Extrusion. To establish the geometric relationships between these primitives, we implement a cu-

rated set of seven distinct constraint types: Coincident, Perpendicular, Parallel, Tangent, Concentric, Intersectant, and Disjoint.

The development of this constraint set was initially informed by the ten constraints proposed in the SketchGraphs (Seff et al., 2020). Given that our work focuses on the four fundamental command types, we first selected the four most prevalent and geometrically significant relationships relevant to them: Coincident, Perpendicular, Parallel, and Tangent. However, we identified a limitation in the SketchGraphs framework, namely the absence of explicit constraints for describing interactions between two circular entities. To address this gap and ensure comprehensive modeling capabilities, we introduced three additional constraints specifically for this purpose: Concentric, Intersectant, and Disjoint. This expanded set enables our framework to represent a richer and more complete spectrum of geometric interactions.

## B.2 LLM-BASED CONSTRAINT ANNOTATION

We employ a large language model for the automated annotation of these curve constraints. The model is architected to perform a sequential analysis, inferring the geometric relationship between each command and its immediate predecessor by examining their respective parameters. For the initial command in a sequence, its relationship with the global origin is determined instead.

This inference is achieved through a deliberate reasoning process. Specifically, we utilized the Gemini-2.5-flash-thinking (Comanici et al., 2025) model for this annotation task. The prompt provided to the model to elicit this analysis is detailed below:

```
**Instruction Goal:** Identify geometric constraints within a CAD
    operation sequence.

**Background Information:**
In CAD modeling, four fundamental atomic operation types are defined:
    [line, arc, circle, extrusion].
There are also seven core geometric constraint types: [Coincident,
    Perpendicular, Parallel, Tangent, Concentric, Intersectant,
    Disjoint].

**Input Data:**
You will be provided with a textual description of a CAD model, its
    atomic operation sequence, and the full parameterized operation
    sequence.
CAD Model Text Description:
{text}
Atomic CAD Operation Sequence:
{atom_cadseq}
Parameterized Operation Sequence:
{cadseq}

**Your Task:**
Please analyze the CAD model description and the provided operation
    sequences carefully to infer the geometric constraints between *
    consecutive* operations.

**Output Format Requirements:**
Output the inferred constraint sequence as a single-line JSON list,
    encapsulated within <constraint> and </constraint> tags. No
    additional explanations or verbose descriptions are needed; **only
     directly output** the final constraint sequence.

**Example Output:**
<constraint>[line, Disjoint, line, Perpendicular, circle, Intersectant
    , circle, Concentric, arc, Tangent, extrusion, Perpendicular]</
    constraint>

**Rules for Constraint Sequence Construction:**
The sequence format is: [action1, constraint1, action2, constraint2,
    ..., action_n, constraint_n]
```

```
Where:
1.  constraint_n represents the geometric relationship between
    action_n and the preceding operation action_n-1.
2.  If an action is the first operation in a sketch, its constraint
    should describe its relationship with the origin.
3.  If an action is extrusion, its constraint should describe its
    relationship with the preceding 2D sketch (or feature).
4.  When multiple constraint possibilities exist, output the
    constraint with the highest priority. Note that 'Coincident' has
    the lowest priority.
```

## C  EXAMPLE OF CAD COMMAND SEQUENCE AND CURVE CONSTRAINT SEQUENCE

This section provides a concrete example to demonstrate the correspondence between a sequence of CAD commands and its associated curve constraints sequence. The example showcases a complete data instance, comprising the initial textual design requirement, the generated CAD command sequence, and the resulting annotated curve constraint sequence, where both the CAD Command Sequence and the Curve Constraint Sequence will be presented in pseudocode. This serves to clarify the relationship between our model's inputs and the target outputs.

**Text Requirement**

```
The CAD model comprises two parts. The first part is a rectangular prism
    featuring a curved top edge. The second part is a cylindrical object
    with flat top and bottom surfaces. These two parts are joined by a
    curved surface.
```

**CAD Command Sequence**

```
Arc(start_point=[0.1102,0.3035], mid_point=[0.0573,0.2709], end_point
    =[0.0035,0.2057])
Arc(start_point=[0.0035,0.2057], mid_point=[-0.0476,0.1472], end_point
    =[-0.1065,0.0902])
Extrusion(depth=0.0902, base_z_height=0.0)
Circle(center=[0.0499,0.3493], radius=0.0501)
Extrusion(depth=0.0541, base_z_height=0.1672)
```

**Curve Constraint Sequence**

```
Arc: Disjoint
Arc: Tangent
Arc: Tangent
Extrusion: Perpendicular
Circle: Concentric
Extrusion: Perpendicular
```

## D  DERIVATION OF THE ATTENTION MASK FUSION EQUIVALENCE

This section provides a formal mathematical derivation of the equivalence between fusing attention masks and coupling the attention mechanisms of a base model with an additional module. This concept was qualitatively introduced in the main text, specifically within the section on Point-Bias Attention Mask. Our primary objective is to rigorously prove that, for every layer in the Transformer architecture, the attention computation performed with fused masks is mathematically identical to that of a fully coupled model.

To begin the proof, we first formally define our key operator. Let $\mathcal{PEO}$ be a zero padding matrix expansion operator that maps representations from a smaller local context to a larger global one, based on the positions of elements $\{p_i\}$ within a global sequence $[S_1, S_2, \cdots, S_N]$. The core of our

proof lies in demonstrating that this operator functions consistently across all layers of the network. This allows us to show that for any decoder layer, the Query, Key, and Value $(Q, K, V)$ matrices of the base model can be approximated as an expansion of those from the additional module. This layer-wise equivalence is the critical step that enables the fusion of attention maps by simply combining their masks.

For simplicity of notation, we represent the attention mechanism as $A = (Q \cdot K) \cdot V$. During autoregressive training, Transformers use attention masks to control information flow. Therefore, the calculation in practice is $A = \mathbf{M}\{Q \cdot K\} \cdot V$. Let $\mathbf{M}$ denote the attention mask matrix.

To distinguish the operations within the two modules, we use the subscript 1 for the base model and 2 for the additional module.

- The inputs to the decoders, which originate from the encoder's output, are denoted as $I_1$ and $I_2$. They are respectively combined with position encodings $\mathbf{PE}_1$ and $\mathbf{PE}_2$ before the first decoder layer.
- These combined inputs are projected into Query, Key, and Value sets: $(Q_1, K_1, V_1)$ for the base model and $(Q_2, K_2, V_2)$ for the additional module.

### D.1 PROPAGATION OF THE EXPANSION PROPERTY THROUGH LAYERS

For the $l^{th}$ decoder layer, the Query, Key, and Value matrices are computed by linearly projecting the output of the preceding layer, $F^{l-1}$, using learnable weight matrices $W_Q$, $W_K$, and $W_V$:

$$Q^l = W_Q \cdot F^{l-1}, \quad K^l = W_K \cdot F^{l-1}, \quad V^l = W_V \cdot F^{l-1}. \tag{4}$$

The input to the first layer ($l = 1$) is the positionally encoded token sequence, $F^0 = \mathbf{PE}\{I\}$.

**Step 1: Base Case for the First Layer ($l = 1$)**

We begin by establishing the base case. When the positional encoding is approximately linear, the $\mathcal{PEO}$ operator can be distributed over the input sequence and the encoding function. To proceed, we establish the core assumption that the position encoding for the base model can be constructed by expanding the position encoding from the additional module into the base model's larger sequence space. This relationship is formally expressed as $\mathbf{PE}_1 = \mathcal{PEO}(\mathbf{PE}_2)$, which leads to:

$$\mathcal{PEO}(\mathbf{PE}_2\{I_2\}) = \mathcal{PEO}(\mathbf{PE}_2)\{\mathcal{PEO}(I_2)\} = \mathbf{PE}_1\{I_1\}. \tag{5}$$

This allows us to relate the Query matrix of the additional module to that of the base model. Assuming the weight matrix $W_{Q_1}$ is a padded version of $W_{Q_2}$, i.e., $W_{Q_1} = \mathcal{PEO}(W_{Q_2})$, we get:

$$\begin{aligned}
\mathcal{PEO}(Q_2^l) &= \mathcal{PEO}(W_{Q_2} \cdot \mathbf{PE}_2\{I_2\}) \\
&= \mathcal{PEO}(W_{Q_2}) \cdot \mathcal{PEO}(\mathbf{PE}_2\{I_2\}) \\
&= W_{Q_1} \cdot \mathbf{PE}_1\{I_1\} \\
&= Q_1^l.
\end{aligned} \tag{6}$$

The same logic applies to Key and Value, establishing the base case for the first layer.

**Step 2: Extension to Subsequent Layers ($l > 1$)**

This property propagates through all subsequent layers. Since linear transformations are compatible with the $\mathcal{PEO}$ operator, the features of the base model consistently remain an expanded version of the additional module's features at every layer, maintaining the relationship $F_1^{l-1} = \mathcal{PEO}(F_2^{l-1})$. Therefore, for any layer $l$, its input $F^{l-1}$ follows the expansion property, and consequently, so do its $Q, K$, and $V$ matrices:

$$\begin{aligned}
\mathcal{PEO}(Q_2^l) &= \mathcal{PEO}(W_{Q_2} \cdot F_2^{l-1}) \\
&= \mathcal{PEO}(W_{Q_2}) \cdot \mathcal{PEO}(F_2^{l-1}) \\
&= W_{Q_1} \cdot F_1^{l-1} \\
&= Q_1^l.
\end{aligned} \tag{7}$$

By induction, this demonstrates that for all layers, the expansion operator $\mathcal{PEO}$ can be applied to the $Q, K$, and $V$ matrices of the additional module to yield those of the base model:

$$\mathcal{PEO}(Q_2^l) = Q_1^l, \quad \mathcal{PEO}(K_2^l) = K_1^l, \quad \mathcal{PEO}(V_2^l) = V_1^l. \tag{8}$$

## D.2 Fusing Attention Maps via Mask Fusion

This recursive property in Eq. equation 8 is powerful because it allows us to fuse the final attention outputs ($A_1$ and $A_2$) by simply fusing their masks. Let us expand the sum of the two attention maps, where $\mathbf{M}_1$ and $\mathbf{M}_2$ are the respective attention masks:

$$
\begin{aligned}
A_1 + \mathcal{PEO}(A_2) &= \mathbf{M}_1\{Q_1 \cdot K_1\} \cdot V_1 + \mathcal{PEO}(\mathbf{M}_2\{Q_2 \cdot K_2\} \cdot V_2) \\
&= \mathbf{M}_1\{Q_1 \cdot K_1\} \cdot V_1 + \mathcal{PEO}(\mathbf{M}_2)\{\mathcal{PEO}(Q_2) \cdot \mathcal{PEO}(K_2)\} \cdot \mathcal{PEO}(V_2) \\
&= \mathbf{M}_1\{Q_1 \cdot K_1\} \cdot V_1 + \mathcal{PEO}(\mathbf{M}_2)\{Q_1 \cdot K_1\} \cdot V_1 \\
&= (\mathbf{M}_1 + \mathcal{PEO}(\mathbf{M}_2))\{Q_1 \cdot K_1\} \cdot V_1.
\end{aligned}
\tag{9}
$$

This result shows that summing the two attention outputs is equivalent to performing a single attention calculation using a fused mask, $\mathbf{M}_3 = \mathbf{M}_1 + \mathcal{PEO}(\mathbf{M}_2)$.

Thus, the derivation is complete. We have shown that coupling the full attention mechanisms is mathematically equivalent to fusing their attention masks, provided the positional encodings are approximately linear. This assumption is valid as CAD command sequences are typically much shorter than those in natural language processing. Consequently, the sinusoidal positional encodings used in our model behave in a nearly linear fashion over these short ranges.

## E    Few-Shot CAD Command Sequence Generation by LLMs

This section details the structured approach of our CAD model generation prompt, which is engineered to guide the LLMs through a precise design process. It is composed of three key sections: the **Input Text for Design Requirements** (providing natural language description), the **Output JSON Format Specification** (rigorously defining the required JSON schema for CAD commands), and an **Example for Text Requirements to JSON Output** (illustrating the translation of requests into structured CAD commands). This integrated methodology ensures accurate and compliant CAD model generation.

```
You are an expert CAD designer and a JSON output generation specialist
    . Your task is to design a 3D CAD model based on the provided
    textual description and output a sequence of CAD commands in a
    strictly defined JSON format.
The CAD model should be composed of one or more parts. Each part will
    have its own coordinate system (defined by Euler Angles for
    rotation and a Translation Vector for position), a 2D sketch (
    which can contain multiple faces and loops), extrusion parameters,
     and a descriptive section.
The sketch can utilize four primitive operations: `line`, `arc`, `
    circle`. The 3D model is constructed using the `extrusion`
    operation.

**Input Text for Design Requirements:**
<text>{text}</text>

**Output JSON Format Specification:**
The output must be a single JSON object, strictly adhering to the
    schema below. It must be wrapped in `<mnjson>` and `</mnjson>`
    tags. Do not include any other text or explanation outside these
    tags.

```json
<mnjson>
{
  "final_name": "string",
  "final_shape": "string",
  "parts": {
    "part_X": {
      "coordinate_system": {
        "Euler Angles": [ "float", "float", "float" ],
        "Translation Vector": [ "float", "float", "float" ]
```

```
918          },
919        "sketch": {
920          "face_X": {
921            "loop_X": {
922              "line_X": {
923                "Start Point": [ "float", "float" ],
                   "End Point": [ "float", "float" ]
924              },
925              "arc_X": {
926                "Start Point": [ "float", "float" ],
927                "Mid Point": [ "float", "float" ],
                   "End Point": [ "float", "float" ]    face.
928              },
929              "circle_X": {
930                "Center": [ "float", "float" ],
931                "Radius": "float"
932              }
933            }
934          }
935        },
936        "extrusion": {
937          "extrude_depth_towards_normal": "float",
938          "extrude_depth_opposite_normal": "float",
939          "sketch_scale": "float",
940          "operation": "string"
941        },
942        "description": {
943          "name": "string",
944          "shape": "string",
945          "length": "float",
          "width": "float",
946          "height": "float"
947        }
948      }
    }
949  }
     </mnjson>

951  **Example for Text Requirements to JSON Output**
952  Example 1
953  <text>...</text>
954  </mnjson>...</mnjson>
```

## F  EVALUATION METRIC FORMULAS

This section details two new metrics proposed for evaluating constraint awareness in CAD generation: Relative Position Consistency (RPC) and Geometric Constraint Consistency (GCC).

### F.1  RELATIVE POSITION CONSISTENCY (RPC)

Relative Position Consistency (RPC) aims to quantify the generated model's ability to maintain original point-level relative relationships while remaining invariant to global translation. It measures the preservation of internal shape structure by computing the average discrepancy between corresponding relative vectors from the ground-truth and generated models. Lower RPC values indicate better preservation of the internal shape structure.

Given:

- $P = \{p_1, p_2, \ldots, p_N\}$: A set of $N$ key points in the ground-truth CAD model, where $p_k \in \mathbb{R}^D$ (typically $D = 2$ or 3).

- $\hat{P} = \{\hat{p}_1, \hat{p}_2, \ldots, \hat{p}_N\}$: A set of $N$ corresponding key points in the generated CAD model, where $\hat{p}_k \in \mathbb{R}^D$.

The RPC is calculated as follows:

$$\text{RPC}(P, \hat{P}) = \frac{1}{\sum_{1 \le i < j \le N} 1} \sum_{1 \le i < j \le N} \left\| (p_j - p_i) - (\hat{p}_j - \hat{p}_i) \right\|_2,$$

where:

- $p_j - p_i$ represents the relative vector between points $p_i$ and $p_j$ in the ground-truth model.
- $\hat{p}_j - \hat{p}_i$ represents the relative vector between corresponding points $\hat{p}_i$ and $\hat{p}_j$ in the generated model.
- $\| \cdot \|_2$ denotes the L2-norm (Euclidean distance) of a vector.
- The sum iterates over all unique pairs of points $(i, j)$ where $1 \le i < j \le N$. The denominator term $\sum_{1 \le i < j \le N} 1$ represents the total number of such unique pairs, which is $\frac{N(N-1)}{2}$.

This metric directly compares the deviation between the original and generated relative vectors. Since relative vectors are inherently independent of global translation, this metric automatically achieves robustness to global translation.

### F.2 GEOMETRIC CONSTRAINT CONSISTENCY (GCC)

Geometric Constraint Consistency (GCC) evaluates the model's adherence to geometric constraints at the curve level. It leverages a large language model to extract constraint sequences from both the generated and ground-truth CAD command streams, and then measures their agreement. Higher GCC values indicate stronger adherence to geometric constraints.

Given:

- $C_G = \{c_{G,1}, c_{G,2}, \ldots, c_{G,M}\}$: The set of constraint sequences extracted from the ground-truth CAD command stream.
- $C_P = \{c_{P,1}, c_{P,2}, \ldots, c_{P,K}\}$: The set of constraint sequences extracted from the generated CAD command stream.

Each constraint $c$ is a standardized representation (e.g., a structured string or object describing its type, affected entities, and parameters) to allow for accurate comparison.

The GCC is calculated as follows, directly computing **Accuracy**:

$$\text{GCC}(C_G, C_P) = \frac{|C_G \cap C_P|}{|C_G|},$$

Where:

- $|C_G|$ is the total number of constraints in the ground-truth constraint set.
- $|C_P|$ is the total number of constraints in the generated constraint set.
- $|C_G \cap C_P|$ is the number of common (i.e., matching) constraints found in both the ground-truth and generated constraint sets.

This simplified GCC metric directly quantifies the proportion of necessary geometric constraints from the ground truth that were successfully captured by the generated model. It measures how many of the expected constraints are correctly identified and present in the generated output.

## G   ADDITIONAL QUALITATIVE EXAMPLE

This section complements the ablation study presented in the main text by offering additional qualitative examples. For each case, we provide the full text requirement alongside the generated output

to clearly illustrate the direct link between the specified requirement and the model's response. The results consistently demonstrate the effectiveness of our proposed components, with our full model (Ours) producing outputs that are notably more structurally complete, faithful to the ground truth (GT), and better aligned with the text requirements compared to the baseline and its variants.

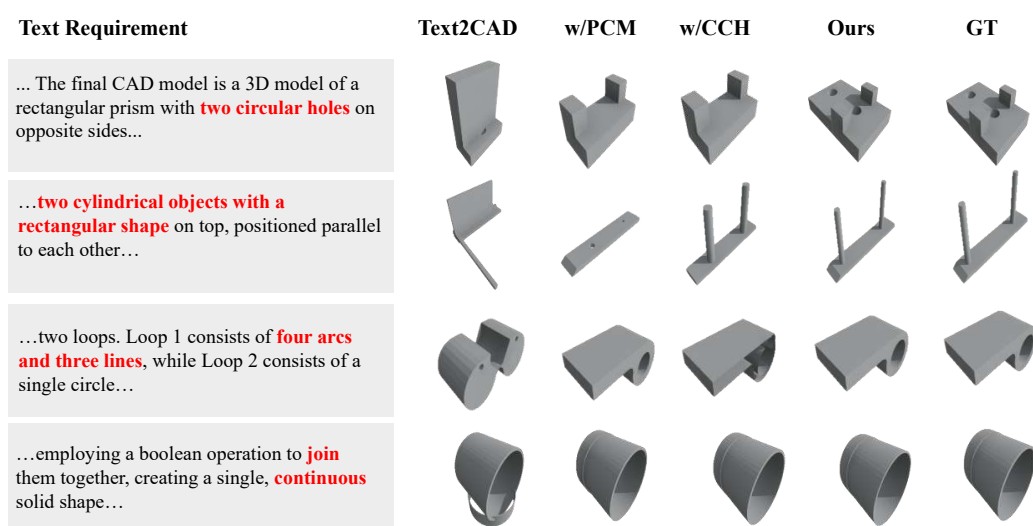

Figure 5: Qualitative Comparison on Models with Component-level Requirements.

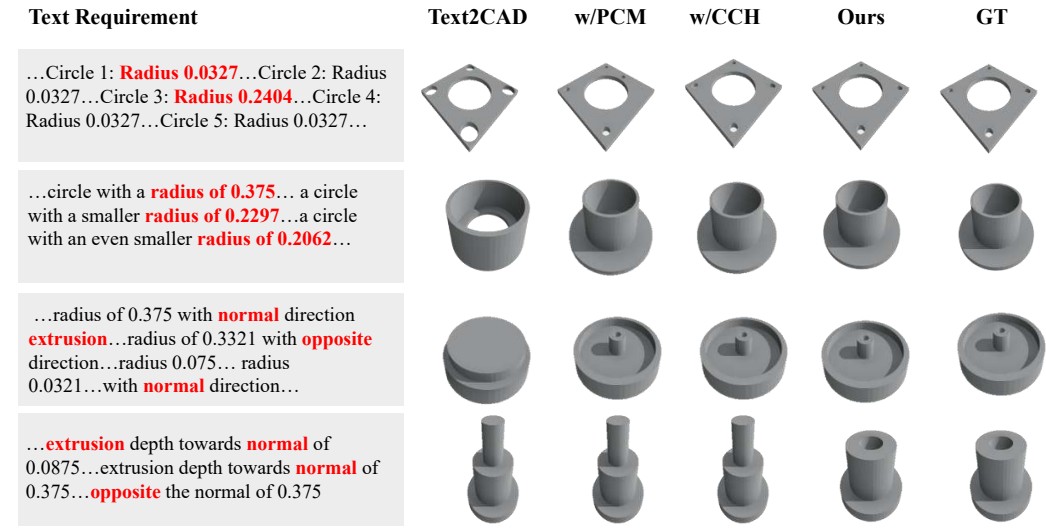

Figure 6: Qualitative Comparison on Models with Accurate Relative Sizing and Positioning

| Text Requirement | Text2CAD | w/PCM | w/CCH | Ours | GT |
|---|---|---|---|---|---|

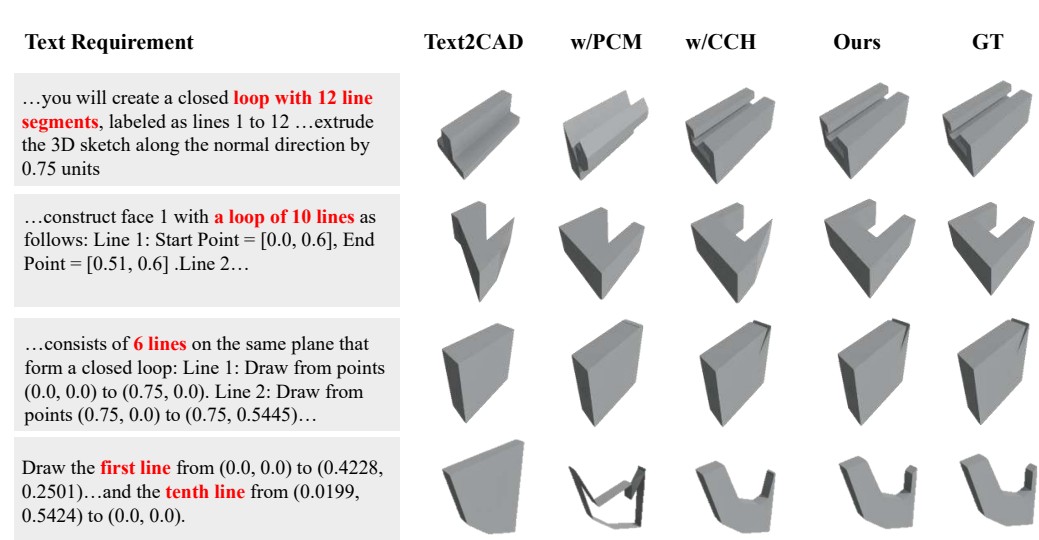

Figure 7: Qualitative Comparison on Models Generated from Complex Polyline-based Sketches.

| Text Requirement | Text2CAD | w/PCM | w/CCH | Ours | GT |
|---|---|---|---|---|---|

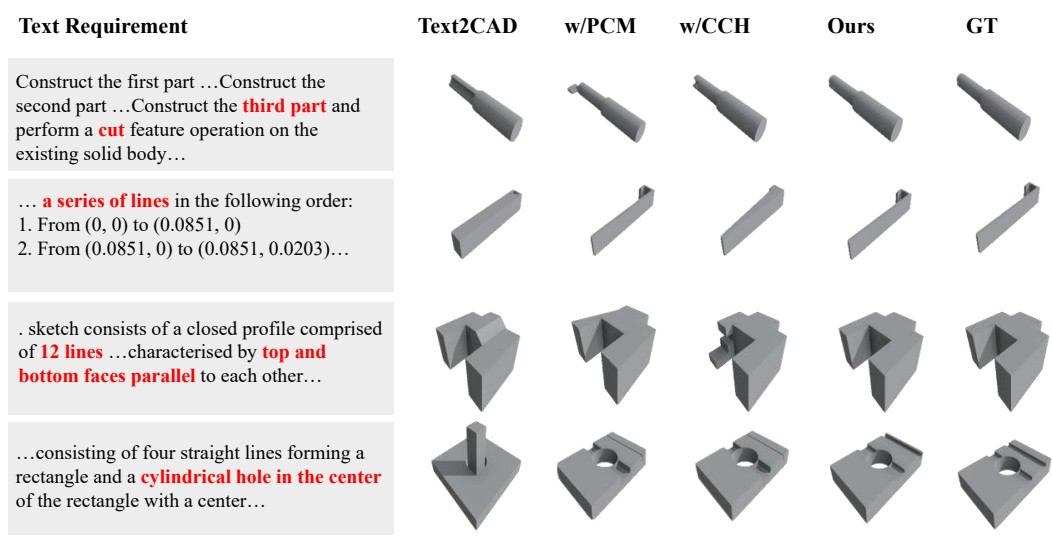

Figure 8: Qualitative Comparison on Models with Other Complex 3D Assets.

## H LLM USAGE

Large Language Models (LLMs) were used to aid in the writing and polishing of the manuscript. Specifically, we used an LLM to assist in refining the language, improving readability, and ensuring clarity in various sections of the paper. The model helped with tasks such as sentence rephrasing, grammar checking, and enhancing the overall flow of the text.

It is important to note that the LLM was not involved in the ideation, research methodology, or experimental design. All research concepts, ideas, and analyses were developed and conducted by the authors. The contributions of the LLM were solely focused on improving the linguistic quality of the paper, with no involvement in the scientific content or data analysis.