# OpenReview forum: "IGeoCAD: Unlocking Intrinsic Geometric Awareness for CAD Command Sequence Generation"
_ICLR.cc/2026/Conference — Submitted to ICLR 2026_

### Official Review · Reviewer_J5ij · 2025-10-20

**Soundness:** 2
**Presentation:** 2
**Contribution:** 2
**Rating:** 2
**Confidence:** 5

**Summary:**

The paper presents a novel text-to-CAD approach called IGeoCAD. Authors claim that both point awareness and curve awareness are important for this conditional CAD generation. To improve the model in these two directions author introduce novel Point Constraint-Aware
Module and the Curve Constraint-Aware Head. The proposed IGeoCAD beats Text2CAD and LLM baselines on Text2CAD benchmark.

**Strengths:**

- The paper is clearly written and logically structured, making the methodology and results straightforward to understand.
- It includes comparisons with five baseline methods on the Text2CAD dataset, demonstrating performance differences across several evaluation metrics.
- The work introduces two new modules -- a Point Constraint-Aware Module and a Curve Constraint-Aware Head -- to incorporate geometric reasoning at both point and curve levels.

**Weaknesses:**

- The proposed method is order of magnitude worse compared to state-of-the-art methods. On Text2CAD (L3) benchmark authors claim to outperform baseline from 29.3 to 21.2 mean Chamfer Distance. However in their related work section they cite methods, with order of magnitude better performance (and open-source code). Namely, Cadrille achieves 3.9 Chamfer Distance score. Other works with significantly better results on this benchmark include Text-to-CadQuery, CAD-Coder, and CADmium.

- Absolute majority of provided metrics are unsound and misleading. Text2CAD paper introduces 4 level of text prompts from L0 (abstract) to L3 (expert). And only L3 is suitable for CAD reconstruction (assuming the single correct prediction). For this end authors of Text2CAD measure F1 and CD scores only on L3 benchmark. L0 prompts include "generate a cylindrical object", and there can not be a single correct answer, as there can be imagined infinite amount of correct cylindrical objects. That is why Text2CAD authors introduce two metrics: GPT4-evaluation and user study evaluation. However, IGeoCAD is still compared to existing approaches with F1 and CD on all benchmarks from L0 to L3. All these ~2e2 CD metrics doesn't show anything relevant, and are misleading for the readers.

- The single evaluation benchmark is not enough to demonstrate the advantages of proposed PCM and CCH modules. Recent papers, including mentioned in related work, Cadrille or CAD-MLLM evaluate their approaches on up to 3 datasets (DeepCAD, Fusion360, CC3D) and 3 different input modalities (images, point clouds, text). However IGeoCAD is only evaluated on DeepCAD with textual prompt modality. I think, the contributions of proposed modules can be shown on much more of the listed datasets and modalities. Even for textual modality there is Omni-CAD dataset from CAD-MLLM paper, or Fusion360 annotated with texts from CADMium paper.

**Questions:**

- How is the proposed IGeoCAD compared to recent (cited by authors) works, including Text-to-CadQuery, CAD-Coder, CADmium, CAD-MLLM?

- What is the point of measuring Chamfer distance between gt and predicted 3D objects for prompts like "generate a cylindrical object"? What does GPT4-evaluation and user study evaluation from the Text2CAD paper show for IGeoCAD?

- What are the advantages of proposed PCM and CCH modules on benchmarks, other than DeepCAD text?

---

### Official Review · Reviewer_NxKz · 2025-10-31

**Soundness:** 3
**Presentation:** 3
**Contribution:** 2
**Rating:** 6
**Confidence:** 3

**Summary:**

This paper addresses CAD command sequence generation by proposing IGeoCAD, a framework that enhances geometric awareness through explicit constraint modeling at two levels: point-level and curve-level. The authors introduce two plug-and-play modules: the Point Constraint-Aware Module (PCM) using point-biased attention masks, and the Curve Constraint-Aware Head (CCH) employing multi-task learning with LLM-generated constraint sequences. Experiments on the Text2CAD dataset show improvements over baselines, with ~20% average reduction in Chamfer Distance across different text granularity levels.

**Strengths:**

1- Well-motivated problem formulation: The decomposition of geometric awareness into point-level and curve-level constraints is intuitive and well-articulated. The distinction between requirement-driven point constraints and procedural curve constraints provides a clear conceptual framework.


2- Novel evaluation metrics: The introduction of RPC (Relative Position Consistency) and GCC (Geometric Constraint Consistency) provides targeted evaluation of point-level and curve-level geometric awareness.

**Weaknesses:**

1- Single-dataset evaluation and no uncertainty estimates; ablation table/terminology inconsistencies.

2- Missing analysis: No failure case analysis and no analysis of which types of geometric constraints benefit most from the approach

3- CD and JSD measure geometric fidelity but don't directly measure constraint satisfaction

**Questions:**

Can you provide quantitative analysis of the quality and consistency of LLM-generated constraint labels?

Can you provide results with statistical significance tests (e.g., multiple runs with different seeds)?

What happens when you use ground-truth constraints (if available) instead of LLM-generated ones?

---

### Official Review · Reviewer_Resf · 2025-10-31

**Soundness:** 2
**Presentation:** 3
**Contribution:** 2
**Rating:** 2
**Confidence:** 4

**Summary:**

The authors aim to address the lack of explicit modeling for point and curve constraints in existing CAD modeling sequence generation methods—particularly those that operate on molding sketches composed of lines, arcs, and splines (similar to the CAD sketches constructed in SolidWorks before executing extrusion commands).

They propose two modules: PCM (Point Constraint-Aware Module) and CCH (Curve Constraint-Aware Head), which handle point-level and curve-level constraints respectively.
They also introduce two new metrics for evaluating point and curve constraint satisfaction.
Both PCM and CCH are designed to be plug-and-play components that can be applied to various CAD sequence generation models, including text-driven generation models.

The key idea of PCM is to increase the attention weights of point tokens, while CCH predicts geometric constraints between adjacent curve primitives.

**Strengths:**

1. This paper exhibits a fair degree of originality, being among the first to focus on point and curve constraints in CAD command sequence generation.

2. The proposed evaluation metrics—RPC and GCC—provide a more interpretable and fine-grained assessment of geometric reasoning, representing a valuable addition to the field.

3. The ablation experiments are sufficiently comprehensive, and the paper includes extensive quantitative evaluations.

**Weaknesses:**

1. Theoretical depth and problem difficulty
The main concern lies in the limited theoretical depth and the relative simplicity of the proposed solutions. In essence, the proposed method focuses on learning geometric constraints in 2D CAD sketches. Such factors have been explored in several prior studies [1, 2], although they have not been applied to the task of CAD command sequence generation.

[1] Ganin Y, Bartunov S, Li Y, et al. Computer-aided design as language[J]. Advances in Neural Information Processing Systems, 2021, 34: 5885-5897.
[2] Seff A, Ovadia Y, Zhou W, et al. Sketchgraphs: A large-scale dataset for modeling relational geometry in computer-aided design[J]. arXiv preprint arXiv:2007.08506, 2020.

2. Writing clarity
The abstract does not clearly describe the task setup—specifically, what type of input data (text, point cloud, or image) is used to generate the modeling sequence.

It was not until line 76 that I realized the “points” and “curves” mentioned refer to entities within the sketch used for extrusion operations (as in SolidWorks). This should be explicitly stated in the abstract to avoid confusion.

Around line 158, it initially appeared that the command sequence representation followed DeepCAD’s style; only after reading the supplementary materials did it become clear that it follows the OpenECAD format. It would be helpful to note this in the main text.

Figure 2 alone makes it difficult to understand the workflow. Readers only realize later that a standard Transformer architecture is used. The figure caption could briefly mention this to improve clarity.

3. Lack of failure case analysis
Although visual examples demonstrate improvement, the paper does not analyze failure cases where IGeoCAD makes incorrect constraint predictions. Including such analysis would clarify the limits of the model’s geometric reasoning capabilities.

**Questions:**

1. Will increasing the attention weights for point tokens in PCM cause the model to ignore other important information?

2. Can the constraints between adjacent curves be computed directly (e.g., by calculating the angle between them to determine perpendicularity) instead of relying on an LLM?

---

### Official Review · Reviewer_4MeP · 2025-11-01

**Soundness:** 3
**Presentation:** 3
**Contribution:** 2
**Rating:** 2
**Confidence:** 4

**Summary:**

IGeoCAD focuses on the task of "CAD command sequence generation", aiming to address the core challenge that existing methods struggle to accurately capture the complex geometric relationships between primitives such as points and curves. The study proposes the "dual geometric awareness" paradigm—point awareness (aligning with high-level design requirements to ensure precise point positioning and consistent constraints) and curve awareness (following the logical flow of commands to maintain global geometric integrity)—and based on this, designs a Transformer framework incorporating a Point Constraint Awareness Module (PCM) and a Curve Constraint Awareness Head (CCH).
In terms of technical approach, IGeoCAD adopts an encoder-decoder Transformer as its backbone: the encoder converts design requirements into adaptive embedding vectors, while the decoder generates command sequences in an autoregressive manner. The PCM enhances point-level dependency modeling through a point bias attention mask, and the CCH optimizes curve-level constraint reasoning by leveraging curve constraint sequences generated by LLMs and dual-head training. For experiments, based on the Text2CAD dataset (four text granularities: L0-L3), IGeoCAD outperforms traditional CAD command generation models (DeepCAD, Text2CAD) and general-purpose large language models (GPT-5, Gemini 2.5 Flash, etc.) across sequence completeness (F1 score) and geometric awareness capabilities (CD, JSD, RPC, GCC). These results validate its effectiveness in end-to-end generation from design requirements to high-precision 3D models.

**Strengths:**

Problem Decomposition and Conceptual Innovation: For the first time, the "geometric awareness" system in CAD command generation is decomposed into point awareness and curve awareness. It clarifies the core distinction that point constraints are driven by high-level requirements while curve constraints are guided by process logic, breaking the limitation of existing methods that "implicitly model geometric relationships" and providing a clear direction for the specialized optimization of geometric constraints.
Lightweight and Practical Module Design: Both the PCM and CCH adopt a "plug-and-play" design that requires no reconstruction of the Transformer backbone. The PCM enhances point-level attention through an attention mask, and the CCH optimizes curve constraints by leveraging LLM-generated constraint sequences and dual-head training.

**Weaknesses:**

Lack of qualitative analysis on the effectiveness of CCH, and the decoupling of the constraint head and command head may affect practical applicationsThis decoupling may lead to the issue where "the constraint head identifies conflicts, but the command head still generates incorrect parameters". For instance, the CCH predicts that a straight line should be tangent to a preceding arc, yet the endpoint coordinates of the straight line generated by the command head still fail to meet the tangency condition. This ultimately impairs the geometric correctness of the 3D model and reduces its executability in practical CAD software. However, the paper does not discuss or verify this potential risk.
Insufficient description of the specific calculation methods for RPC and GCC metricsThe paper only defines RPC as "calculating the average difference of corresponding relative vectors, with lower values being better" and GCC as "extracting constraint sequences via LLMs and calculating matching degree, with higher values being better", without specifying key details. The lack of metric calculation details makes it difficult for other studies to reproduce the experimental results and to evaluate the rationality of the metrics themselves.
Doubts about the validity of small value improvements after multiplying the CD metric by 10³
Inadequate detailed presentation of Bad Cases, and unclear specific shortcomings of existing methods under the L3 granularityThe paper only mentions in the qualitative analysis that "baseline models suffer from missing components and structural breaks", but fails to supplement Bad Case details for the L3 granularity. For example, when an L3 requirement explicitly states "draw a straight line from (0.0, 0.0) to (0.4228, 0.2501)", what incorrect parameters would Text2CAD or LLMs generate (e.g., an endpoint coordinate deviation of 0.01, or directly generating the wrong command type), and whether the errors are concentrated in "point positioning deviations" or "curve constraint conflicts". Additionally, the paper does not explain "why existing methods still have shortcomings under the L3 granularity". For instance, Text2CAD achieves a CD of 29.30×10³ at the L3 granularity—whether the specific error source is incorrect parameter calculation or unmet constraints—leaving readers unable to understand "the core flaws of existing methods under detailed descriptions".

**Questions:**

See weakness

---

### Meta-Review · Area_Chair_4HGw · 2026-01-07

**Summary:**

* Limited technical depth and incremental contribution.
* Misleading and unsound evaluation metrics.
* Weak empirical comparison to state of the art.
* Over-reliance on LLM-generated constraints.
No rebuttal is provided. Given the ratings of 2,2,6,2, this submission should be rejected.

**Reviewer Concerns:**

No rebuttal is provided.

**Reviewer Scores:**

No rebuttal is provided. Scores will not change.

---

### Decision · Program_Chairs · 2026-01-26

Reject